

# Enhancing fraud detection in auto insurance and credit card transactions: a novel approach integrating CNNs and machine learning algorithms

Ruixing Ming[1,*], Osama Abdelrahman[1,*], Nisreen Innab[2] and Mohamed Hanafy Kotb Ibrahim[3]

[1] School of Statistics and Mathematics, Zhejiang Gongshang University, Hangzhou, China
[2] Department of Computer Science and Information Systems, College of Applied Sciences, AlMaarefa University, Riyadh, Saudi Arabia
[3] Department of Statistics, Mathematics, and Insurance, Faculty of Commerce, Assiut University, Asyut, Egypt
[*] These authors contributed equally to this work.

## ABSTRACT

Fraudulent activities especially in auto insurance and credit card transactions impose significant financial losses on businesses and individuals. To overcome this issue, we propose a novel approach for fraud detection, combining convolutional neural networks (CNNs) with support vector machine (SVM), k nearest neighbor (KNN), naive Bayes (NB), and decision tree (DT) algorithms. The core of this methodology lies in utilizing the deep features extracted from the CNNs as inputs to various machine learning models, thus significantly contributing to the enhancement of fraud detection accuracy and efficiency. Our results demonstrate superior performance compared to previous studies, highlighting our model's potential for widespread adoption in combating fraudulent activities.

Corresponding author
Osama Abdelrahman,
osamamohamad764@gmail.com

# INTRODUCTION

Fraud is a pervasive and global issue with severe consequences for businesses and individuals (*Craja, Kim & Lessmann, 2020*). Detecting and preventing fraudulent activities is crucial in today's highly competitive commerce landscape, as it safeguards companies and investors (*Shi & Zhao, 2023*). Financial fraud poses a significant challenge to the financial sector, necessitating efforts across various industries. These fraudulent schemes, such as insurance fraud and sophisticated skimming techniques, are designed to secure illicit financial gains (*Cheah, Yang & Lee, 2023*). According to the Canadian Anti-Fraud Centre (CAFC) report in 2022, there were over 91,190 fraud reports, with victims suffering losses exceeding $531 million (*Motie & Raahemi, 2023*). This underscores the substantial monetary losses and erosion of trust in financial institutions caused by financial fraud.

Financial fraud encompasses a wide range of fraudulent activities, such as credit card fraud, securities and commodities fraud, fraudulent financial statements, mortgage fraud,

insurance fraud, bank fraud, *etc.* (*Cheng & Cai, 2023*). The two most common financial fraud losses are credit card and insurance fraud. Accordingly, this study focuses specifically on auto insurance fraud, which is a subset of insurance fraud. This choice is made due to the significant portion of expenses incurred by property insurance companies that can be attributed to auto insurance fraud, as well as credit card fraud, given their prevalence and consequential impact. First, the insurance industry suffers billion-dollar losses every year as a result of fraud, according to reports published in 2019 by Insurance Europe and the Association of British Insurers (*ABI, 2021*). Such losses impact insurance companies and escalate costs for honest policyholders through increased premiums. Detecting fraud results in significant savings and reinforces the market's integrity, acting as a strong deterrent against malpractices. In the vast realm of insurance, auto insurance fraud detection emerges as a peculiar challenge, emphasizing the need for innovative solutions. Undeniably, the annual economic impact resulting from auto insurance fraud is anticipated to surpass the staggering sum of $40 billion, as substantiated by the authoritative findings of the Federal Bureau of Investigation in 2022. An auto insurance fraud profoundly impacts property insurance companies, affecting their pricing strategies and long-term socioeconomic advantages (*Maiano et al., 2023*).

Furthermore, fraud leads to immediate financial losses and hinders the growth and advancement of insurance companies. The increasing frequency of fraudulent claims diminishes the solvency of these companies, depleting the reserves allocated for legitimate claim settlements and hampering their ability to provide financial support for new business endeavors. Traditionally, insurance fraud detection mainly relied on manual auditing and expert inspection. However, this approach proved to be expensive, time-consuming, and ineffective. Moreover, it became crucial to identify fraud cases before claim payments were made to minimize financial losses.

Consequently, the detection of auto insurance fraud is of utmost significance as it empowers insurance companies to efficiently manage costs and mitigate the impact of fraudulent activities (*Wang & Xu, 2018*). For these reasons, there is a need to employ artificial intelligence (AI) algorithms to enhance financial fraud detection accuracy and decrease losses. By employing AI algorithms, the detection procedure can be automated, leading to improved efficiency and structure in identifying fraudulent activities with enhanced precision.

The second common fraud is credit card fraud. In recent years, there has been a prominent surge in the usage of credit cards, primarily driven by the rapid growth of e-commerce activities. This escalating reliance on credit card usage has consequently fostered a persistent and continual rise in fraudulent transactions (*Esenogho et al., 2022*). Although credit cards offer numerous advantages to users, they are also linked to issues such as security and fraud. Credit card fraud poses a significant concern for banks and financial organizations. Credit card fraud is the act of unauthorized individuals utilizing credit cards to obtain money or property through deceitful methods. Credit card information is vulnerable to theft through insecure online platforms and web pages. They can also be acquired through identity theft techniques. Scammers can illicitly obtain the credit and debit card numbers of consumers without their authorization or awareness (*Bin Sulaiman,*

*Schetinin & Sant, 2022*). With the evolving tactics of fraudsters, the industry requires sophisticated and transparent methods. This is where AI comes into the picture. With its potential for detecting intricate patterns in vast datasets, AI is poised as a promising tool to address the challenge of insurance fraud detection. This study proposes a new algorithm to improve fraud detection accuracy and uses an oversampling method to handle the imbalanced data.

In recent times, there has been a discernible emphasis on utilizing machine learning techniques to elucidate fraudulent patterns, reflecting a growing awareness of the importance of such approaches in the field of fraud detection. Previous research in fraud detection primarily concentrated on classification approaches, including logistic regression, support vector machines (SVM), bayesian belief networks, and neural networks. Although these conventional methods are widely employed, these algorithms do not adequately address the challenges posed by imbalanced data and the need for frequent collection of fraud activities and periodic relearning. Interestingly, multiple studies have demonstrated that logistic regression, SVM, and random forests exhibit notably superior performance in accurately identifying legitimate transactions compared to fraudulent ones.

Despite the recent advancement in the fraud detection field, a significant research gap still exists regarding the effective handling of imbalanced data in the context of claims fraud detection. Specifically, the current body of literature lacks comprehensive investigations that integrate convolutional neural networks (CNNs) with machine learning algorithms to tackle the unique challenges presented by imbalanced datasets. To address this gap, our proposed research endeavors to introduce an innovative model that capitalizes on the deep features extracted from CNNs, with the aim of augmenting the accuracy of fraud detection while simultaneously enhancing efficiency. Consequently, this study proposes an innovative model for identifying fraud in the auto insurance sector and credit card domains. To address the challenge of imbalanced data, an adaptive oversampling technique is employed, aiming to mitigate the inherent data imbalance issue and uses deep features from CNNs as inputs to machine learning models represent a significant advancement in the field, offering enhanced detection capabilities while maintaining the necessary transparency for practical application in various sectors, including finance and cybersecurity. The motivation behind our study is to provide a comprehensive and nuanced fraud detection systems for auto insurance and credit card transactions. Furthermore, our goal is to demonstrate the better performance of our model over current approaches by means of an extensive evaluation using real-world datasets.

Enhancing the precision of detecting car insurance fraud and credit card is of utmost importance as it will mitigate the financial losses incurred from fraudulent activities. This, in turn, will have a favorable impact on the insurance industry, policyholders, banks and the overall economy. Furthermore, it will aid insurance companies in anticipating fraudulent claims, which will significantly impact their revenue and the satisfaction of their customers.

Concretely, this study seeks to make valuable contributions to the existing scholarly literature to detect fraud in several ways. In terms of methodology, the methodology introduces an advanced feature extraction approach using optimized CNNs that capture patterns and anomalies in transactional data. Integrating new CNN-based feature extraction

with diverse machine learning models, it creates a robust fraud detection system superior existing methods. Furthermore, combining CNNs and machine learning sets a new benchmark for detecting intricate fraud schemes. The model is adaptable and scalable across various fraud scenarios, maintaining high accuracy and low false positives, as validated by rigorous testing with real-world fraud datasets.

In terms of practical value, our model significantly enhances fraud detection for stakeholders in the insurance industry, banks, investors, and policyholders. Integrating our proposed model into existing systems reduces financial losses and improves security for financial institutions and vehicle insurance firms. The model's precision also abridges fraud investigation, optimizing resource allocation and reducing costs. Furthermore, the model's sophisticated features enhance fraud detection systems, providing benefits like improved fraud protection, increased effectiveness, reduced expenses, and elevated trust in specific industries. Moreover, it also proposals transparent insights into decision-making, balancing complexity and interpretability to foster consumer trust. In terms of results, the results of our study demonstrate that the proposed approach displays exceptional performance in comparison to previous studies across auto insurance fraud detection and credit card fraud detection. Additionally, the model's high accuracy and low false positives validate its reliability and suitability for practical implementation. Moreover, these results reinforce our model's practical value and impact in enhancing fraud prevention, improving efficiency, and instilling trust in fraud detection systems. Our proposed model achieved the highest accuracy, precision, recall, F1-score, and AUC in vehicle insurance fraud detection, with values of 98.6%, 100%, 97.72, 98.85, and 96.5, respectively. On the other hand, our suggested model achieved even higher performance, with the highest accuracy, precision, recall, F1-score, and AUC of 99.99%, 100%, 99.97, 99.99, and 100%, respectively, in vehicle insurance fraud detection.

The remainder of the article is organized as follows. In section 'Literature Review', we showcase the literature review. Section 'Methodology' shows the proposed model. In 'Proposed Method', we present the results. Section 'Results and Discussion of the Proposed CNN Model and Ensemble ML Classifier' illustrates the conclusion and future works.

## LITERATURE REVIEW

This section presents a comprehensive review of existing research in auto insurance and credit card fraud detection, focusing on utilizing machine learning and deep learning algorithms.

The considerable financial losses resulting from fraudulent practices have prompted researchers and scholars to strive to develop a robust framework to detect and prevent fraud effectively. Insurance claim fraud represents an intricate and multifaceted phenomenon, often characterized by substantial time and cost requirements for its detection (*Debener, Heinke & Kriebel, 2023*). Therefore, there is a need to employ machine learning and deep learning to address this issue, and leveraging artificial intelligence for enhanced fraud detection can serve as a strong deterrent against fraudulent activities, offering advantages to insurance companies and their loyal policyholders. In light of this, a prominent area of

scholarly research endeavors to improve the efficiency of insurance claim fraud detection and credit card fraud through statistical methods and also to handle imbalanced data by employing different techniques. To enhance the efficiency of fraud detection, Artificial intelligence

In the realm of fraud detection, comprehensive studies of the literature have been done to investigate several approaches that try to enhance accuracy and deal with the problem of unbalanced-data. Scholars have acknowledged the need for precise fraud detection in defending companies and lessening the effects of fraudulent activity (*Craja, Kim & Lessmann, 2020*). Numerous research endeavors have concentrated on employing diverse techniques to augment the comprehensive efficacy of fraud detection algorithms.

## Vehicle insurance fraud detection

Numerous studies have traditionally utilized conventional linear techniques to identify fraudulent insurance claims. However, the application of machine learning techniques in this domain has gained significant popularity. Recent research suggests that leveraging machine learning will play a crucial role in developing more effective fraud detection systems. As a result, several scholars have focused on exploring various techniques to enhance fraud detection performance and tackle the issue of class imbalance. They have evaluated the predictive capabilities of these techniques through comprehensive comparative analyses.

In the study by *Wang & Xu (2018)*, a novel approach was proposed for the detection of auto insurance fraud. The methodology employs a deep learning model integrated with Latent Dirichlet Allocation (LDA)-based text analytics. Deep neural networks are trained to enable accurate fraud detection by extracting text features from accident descriptions in insurance claims using LDA and combining them with traditional numeric features. The experimental results demonstrate that the proposed model achieves the highest accuracy rate of 91.4% compared to SVM and RF algorithms.

The study proposed by *Subudhi & Panigrahi (2020)* presented a distinctive hybrid approach that integrates various supervised classifier models with a genetic algorithm (GA)-based fuzzy C-means (FCM) clustering technique to identify fraudulent activities in vehicle insurance claims. According to the results, SVM performed better than any other classifier, having the best accuracy of 88.45%, specificity, and 83.21% sensitivity. In a similar vein, *Majeed, Abdullah & Mushtaq (2021)* enhanced the existing study by proposing a fuzzy clustering technique that synergistically harnesses the capabilities of both the modified whale optimization algorithm (MWOA) and FCM algorithm. The findings showed that the new method generated an accuracy of 96.25%.

Motivated by the need to achieve even higher accuracy rates, researchers explored alternative algorithms and optimization techniques. *Salmi & Atif (2021)* proposed a new method for detecting fraudulent claims using data mining by utilizing two sample techniques (SMOTE and ROSE) to eliminate the class disparity and test two distinct feature subsets. The results illustrated that random forest performs better than logistic regression with a 95.24% accuracy.

Similarly, *Lv et al. (2022)* developed a resilient model for auto insurance fraud detection was constructed using a Logistic-SVM approach, and its performance was compared against conventional algorithms, including logistic regression and SVM. The outcome of the evaluation demonstrated that the Logistic-SVM model exhibited superior performance, achieving the highest accuracy rate of 96.1% and outperforming the other algorithms. Further this study by *Xia, Zhou & Zhang (2022)* introduced a novel deep-learning framework for the identification of auto insurance fraud. The proposed approach integrates CNN, long- and short-term memory (LSTM), and deep neural network (DNN) architectures to leverage their respective strengths. The results demonstrate that the proposed CNN-LSTM model surpasses previous deep learning models, exhibiting improved accuracy, recall rate, and precision with values of 89.6%, 90.7%, and 89.6%, respectively.

*Aslam et al. (2022)* employed three predictive models, namely naïve Bayes, SVM, and logistic regression, to detect vehicle insurance fraud. Among these models, SVM exhibited the highest accuracy and specificity scores, reaching 94% and 99.77%, respectively. Conversely, the naïve Bayes (NB) algorithm demonstrated the highest precision score of 23.07.

Parallel to these advancements, in this study by *Supriya et al. (2023)*, a hybrid approach is suggested that integrates federated learning (FL), the genetic algorithm (GA), and particle swarm optimization (PSO) to enhance the detection of automobile insurance fraud. The results indicate that the proposed hybrid model achieves the highest accuracy of 94.47%. In a recent study, *Abakarim, Lahby & Attioui (2023)* introduced an innovative approach to address the challenge of imbalanced data through the utilization of analysis-based techniques. The study incorporates three pre-trained CNN models, namely AlexNet, InceptionV3, and Resnet101, which undergo a streamlined process by removing redundant layers. These CNN models are subsequently integrated in parallel with a novel 1D CNN model using Bagged Ensemble Learning, facilitating an effective solution for imbalanced data classification. In this approach, a SVM classifier is utilized to extract individual predictions from each CNN model. Subsequently, the outcomes of these models are combined using the majority polling technique, resulting in a consolidated prediction. The best performance accuracy for the proposed model was 98%. Correspondingly, A predictive model using machine learning techniques was developed by *Okagbue & Oyewole (2023)* to improve the fraud detection accuracy of *Abakarim, Lahby & Attioui (2023)* method and identify potential instances of fraudulent vehicle insurance claims. This study evaluated six algorithms SVM, ANN, AdaBoost, XGboostNB, LR, DT, and RF the best result was 98.5% for Random forest algorithms.

Moreover, *Maina, Moso & Gikunda (2023)* introduced a new method to deal with imbalanced data by employing the oversampling method and the proposed model is based on the XGBoost algorithm to improve fraud detection efficiency. The findings indicated that XGBoost demonstrates strong performance when applied with SMOTE to address imbalanced training datasets, outperforming other algorithms in the study.

## Credit card fraud detection

Recently, credit cards have become a prevalent means of purchasing essential goods that individuals may not be able to afford immediately. However, with the increasing use of credit cards, fraud incidents are also on the rise. Therefore, there is a pressing need to develop a robust and accurate model that can effectively predict and detect credit card fraud (*Singh et al., 2023*). With the rise of big data and the imbalance of data is a significant issue. The imbalance of data is one of the inevitable problems in business, which will affect the performance of classification models.

Prior research on credit card theft revealed that K-nearest neighbor (KNN) outperformed naïve Bayes (NB), RF, DT, SVM, J48, and binary classification technique (BCT) (*Zhao & Guan, 2023*).

The harmful impact of imbalanced data structures on various classification models, including decision trees, artificial neural networks (ANNs), and SVMs, has been widely recognized in academic literature (*Ren et al., 2023*). As a result, a considerable number of researchers have dedicated their efforts to the development of techniques for addressing imbalanced data by employing various methods, which can be categorized into three main approaches: data-level, feature-level, and algorithm-level.

Researchers and practitioners have proposed numerous approaches to address the challenge of identifying fraudulent transactions and overcoming the issue of imbalanced data. Various machine learning techniques have enhanced fraudulent transaction detection accuracy, including supervised and unsupervised methods such as SVM, random forest, isolation forest, local outlier factor, autoencoder, and others. The primary objective is to differentiate between fraudulent and legitimate transactions. Many researchers have extensively examined different approaches and conducted thorough evaluations to determine and compare their predictive prowess. To detect fraudulent activities and address the imbalance data issue, *Fiore et al. (2019)* introduced a novel voting mechanism based on artificial neural networks and an ensemble model that utilizes deep recurrent neural networks to model sequential data and overcome the unbalanced data. The results obtained from the study showcased the impressive performance of their proposed model, with a precision rate of 97.48% and a recall rate of 77.95%, underscoring its effectiveness.

This performance was improved by *Benchaji et al. (2021)* by merging the capabilities of three sub-methods. The first sub-method, the uniform manifold approximation and projection (UMAP), was employed to identify the most informative predictive features. The second sub-method involved utilizing long short-term memory (LSTM) networks to incorporate transaction sequences. Lastly, the attention mechanism was utilized to further enhance the performance of the LSTM networks. Similar in vein, *Esenogho et al. (2022)* presented a new method to improve credit card detection accuracy and also deal with imbalance data by using a long short-term memory (LSTM) method for identifying credit card fraud by employing a neural network ensemble classifier in combination with a hybrid data resampling approach. This new method outperformed sensitivity by 99.6% and specificity by 99.8%. Parallel to these advancements, *Sharma, Singh & Chandra (2022)* suggested an innovative two-phase oversampling technique that combines knowledge transfer, leveraging the synergistic benefits of the Synthetic Minority Over-sampling

Technique (SMOTE) and generative adversarial network (GAN). The results showed the best performance for the SMOTEified technique with a precision of 85% and recall of 80%.

*Forough & Momtazi (2021)* suggested an ensemble model that utilizes deep recurrent neural networks and artificial neural network-based voting mechanisms to identify fraudulent actions by analyzing data sequences. The results clearly indicate that the suggested model surpasses the current leading models in all evaluation categories. Further, *Cheah, Yang & Lee (2023)* proposed a new approach to address the issue of class imbalance effectively and increase fraud detection accuracy. This approach is SMOTE+GAN and GANified-SMOTE. The findings reported that the GANified-SMOTE model achieved the highest precision by 94% and recall by 85%. A novel method called Deep Boosting Decision Trees (DBDT) has been proposed for detecting fraud by *Xu et al. (2023)* which combined the power of gradient boosting and neural networks to enhance fraud detection capabilities. The proposed method outperformed the best performance by AUC 80%. In *Lei et al. (2023)*, the authors introduced a distributed neural network model (DDNN) designed for credit card fraud detection. The model leverages a model optimization algorithm to federate credit card transaction data. The results showed that the accuracy of this model is higher than previous studies by 99.94%.

Moreover, *Fanai & Abbasimehr (2023)* introduced a two-stage framework to enhance fraud detection accuracy. It employs a deep Autoencoder (AE) for dimensionality reduction and three deep learning classifiers: DNN, RNN, and CNN-RNN. Bayesian optimization optimizes the models' hyper parameters. The empirical results highlighted the superiority of the proposed approach in enhancing performance. AE-DNN outperformed the baseline DNN model across all evaluation metrics. Additionally, AE-RNN and AE-CNN_RNN surpass their respective baselines, achieving impressive results with an F score of 83%, Recall of 73%, and precision of 97.8%.

*Van Belle, Baesens & De Weerdt (2023)* introduced CATCHM, an innovative credit card fraud detection method that utilizes representation learning (RL) based on network analysis. This study demonstrates that CATCHM surpasses existing state-of-the-art methods. This outcome substantiates the practical significance of this approach for the industry.

The study by *Leevy, Hancock & Khoshgoftaar (2023)* conducted an evaluation of binary and one-class classification techniques for credit card fraud detection. Five binary-class classification (BCC) learners and three one-class classification (OCC) learners were assessed using performance metrics such as AUPRC (Area Under Precision-Recall Curve) and AUC (Area Under the ROC Curve). The results indicated that binary classification approaches outperformed one-class classification methods in detecting credit card fraud. The BCC learners achieved AUPRC scores ranging from 85.7% to 74.90%, and CatBoost exhibited the highest AUPRC score of 85.67%.

*Zhao & Guan (2023)* introduced CTCN, a method for detecting credit card fraud that utilizes a combination of CTGAN and TCN to handle the problem of imbalanced data effectively. This study enhanced CTGAN by including NCL to address the issue of class overlap in imbalanced datasets. The results demonstrated that the proposed approach attained exceptional performance with regards to Recall by 82.99%, F1-Score by 81.87, and AUC-ROC by 91.47%.

On other hand, some studies used CNN and machine learning. For instance, *Alharbi et al. (2022)* aims to enhance the credit card fraud detection uses a text2IMG conversion technique to transform transaction data into images, and then applies a 16-layer CNN architecture for fraud classification. The text2IMG conversion method proposed in the attached article to represent transaction data as images is a novel aspect of their work. The findings performed that the proposed model achieved the highest accuracy of 99.87%. Moreover, *Mizher & Nassif (2023)* presented a novel approach for credit card fraud detection using a CNN in conjunction with two machine learning algorithms. The results confirmed that the Random Forest classifier achieved the highest accuracy of 99.78% in detecting credit card fraud on the highly imbalanced dataset.

Even though the existing literature on fraud detection methods, there is a noticeable research gap in comparing the results of traditional methods with more advanced techniques. Most studies only compare their findings with a limited range of traditional methods such as SMOTE, hybrid models of GAN with SMOTE, and deep autoencoder neural networks. By expanding the scope of comparison to include more advanced oversampling techniques and ensemble methods, deeper insights can be gained into the effectiveness of different approaches.

Furthermore, the literature primarily evaluates shallow machine learning models like SVM and logistic regression or explores integrating different neural network architectures such as RNN and LSTM with CNN. However, comparing these approaches with more advanced ensemble methods, such as combining ensemble machine learning algorithms like KNN and DT with a CNN model, could provide a more comprehensive understanding of fraud detection techniques. Moreover, while some studies have utilized CNNs for fraud detection, there is a need to improve the feature extraction techniques to enhance model performance, particularly for complex datasets. Feature extraction assumes a pivotal role in converting raw data into a more suitable format for machine learning algorithms. Exploring sophisticated techniques for feature extraction can bridge this gap and make significant contributions to enhancing the overall performance of fraud detection models.

Moreover, the implementation of sophisticated financial detection methods effectively accomplishes the necessary objective. Nevertheless, machine learning and deep learning approaches have effectively tackled these issues on a significant magnitude in recent years. Therefore, there is a need for enhancement in these techniques with regards to the identification of unusual attack patterns, velocity estimates, and big data analysis. Hence, this study aims to introduce a novel algorithm that enhances the accuracy of fraud detection. This will be achieved by developing a CNN architecture capable of extracting intricate and profound features that indicate fraudulent activities. The proposed approach establishes a robust foundation for analysis by identifying and capturing complex patterns and anomalies. Consequently, the extracted intricate features will be integrated into advanced machine learning models, effectively leveraging the benefits of both deep learning and traditional machine learning techniques.

The importance of the suggested CNN model for fraud detection can be explained by the following observations:

Firstly, CNN models play a crucial role in fraud detection, especially in credit card fraud, because they are highly skilled at automatically extracting spatial and temporal characteristics. This capability allows them to perceive intricate patterns in transaction data that may not be immediately evident to human analysts or conventional machine learning methods. The suggested CNN model utilizes this advantage to improve the accuracy of fraud detection by capturing complex features and subtle patterns that are indicative of fraudulent actions.

Furthermore, CNNs in fraud detection applications have the ability to immediately learn features from raw data, which leads to a reduced need for feature engineering. This feature greatly lowers the need for domain specialists to perform intensive feature engineering, resulting in CNN models that are extremely adaptable to different fraud detection scenarios. The model provided in the study showcases this benefit, illustrating its potential use not just in detecting credit card fraud but also in potentially detecting insurance fraud.

Moreover, the incorporation of CNNs with ensemble machine learning techniques introduces an innovative method that greatly enhances the performance of fraud detection. The suggested model can provide more precise and dependable fraud detection compared to conventional methods by utilizing deep features derived by CNNs. The importance of CNNs in improving the prediction skills of downstream models is emphasized by this integrated approach.

The usefulness of the suggested CNN model is confirmed by its excellent accuracy, precision, recall, and AUC measures, especially in the field of credit card fraud detection. The performance indicators emphasize the model's dependability and its ability to minimize incorrect identifications of fraud, which is a critical factor in real fraud detection systems where the expense of false alarms can be significant. Lastly, the ability to adapt across several domains: in addition to its effectiveness in detecting credit card fraud, the suggested CNN model's versatility in detecting other types of fraud, such as insurance fraud, demonstrates its wider usefulness as a generic fraud detection method. The model's adaptability highlights its importance, providing a strong and adaptable solution that can be customised to address various fraud detection jobs (*Khedgaonkar, Singh & Raghuwanshi, 2021*; *Sindi et al., 2021*).

## METHODOLOGY

In this section, we comprehensively examine the dataset used for classifying instances as fraudulent or non-fraudulent. Furthermore, we delve into the application of several machine learning algorithms, namely SVM, KNN, NB, DT, and the CNNs model, which have been implemented as part of our proposed methodology. We present a thorough elucidation and visual depiction of the methodology, with specific emphasis on the proposed CNN model, as illustrated in Fig. 1. The procedures delineated in the block diagram will be expounded upon in greater detail in the subsequent section.

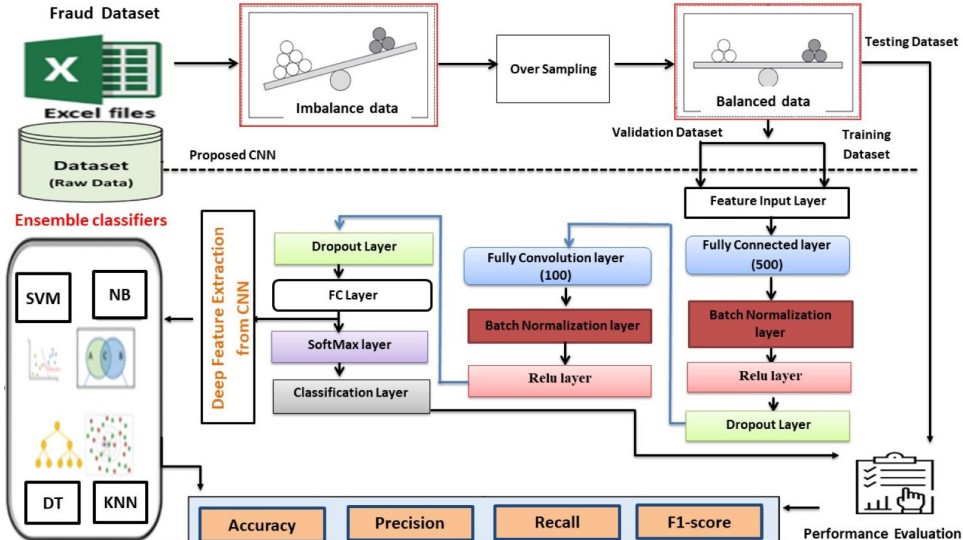

**Figure 1   Diagram illustrating the suggested approach.**

## Data description

To investigate the techniques utilized by state-of-the-art models in detecting occurrences of credit card fraud and auto insurance claim fraud, our analysis employs two real-world datasets obtained from Kaggle, namely vehicle insurance (https://www.kaggle.com/datasets/shivamb/vehicle-claim-fraud-detection) and European credit card transactions, for the purpose of fraud detection. The datasets in this study display an inherent class imbalance, with a notable disproportion between the number of fraudulent and non-fraudulent instances (https://www.kaggle.com/mlg-ulb/creditcardfraud/home). The summary statistics for these datasets are presented in Table 1.

Auto insurance and credit card fraud were specifically chosen as the focus areas for this study due to their significant economic impact and prevalence compared to other forms of fraud. Consider the following evidence that underscores the importance of these two domains: first, Auto insurance fraud is a major problem costing the industry billions of dollars annually. The Insurance Information Institute estimates that auto insurance fraud adds $5.6–$7.7 billion in excess payments to auto insurance claims each year in the US alone. This translates to higher premiums for all policyholders. The pervasiveness and substantial financial toll make auto insurance fraud a critical area demanding advanced detection techniques. Second, credit card fraud is one of the most common types of identity theft. According to the Federal Trade Commission's 2021 Consumer Sentinel Network report, credit card fraud was the most reported type of identity theft, constituting over 390,000 reports. The high volume and direct monetary losses to both consumers and financial institutions underscore the need for robust credit card fraud detection methods.

The techniques used in auto insurance and credit card fraud, such as falsified claims, identity theft, and transaction manipulation, are representative of strategies employed across many other fraud domains. Therefore, developing effective detection models for

**Table 1  Summary statistics of the two datasets.**

| Dataset | Samples | Features | Ratio of minority | Number of Fraud | Number of non-Fraud | Oversampling for Fraud dataset |
|---|---|---|---|---|---|---|
| Vehicle Insurance Fraud | 15,420 | 32 | 5.99% | 923 | 14,497 | 13,574 |
| European credit card transactions Fraud | 284,807 | 30 | 0.173% | 492 | 284,315 | 283,823 |

these two areas can provide insights and methodologies applicable to combating fraud in other sectors as well (https://www.michigan.gov/difs/consumers/fraud/what-is-auto-insurance-fraud). Moreover, From a data perspective, auto insurance and credit card transactions generate vast amounts of structured data suitable for machine learning model development. The availability and quality of data in these domains make them conducive to researching advanced fraud detection techniques like the CNN-based approach proposed in this study (https://www.bankrate.com/insurance/car/fraud/).

Furthermore, the financial services and insurance industries have been at the forefront of adopting machine learning for fraud detection due to the high stakes involved. Focusing on auto insurance and credit card fraud aligns with the sectors most likely to benefit from and implement the novel techniques presented in this research (https://www.chargebackgurus.com/blog/auto-insurance-chargebacks). Lastly, the selection of auto insurance and credit card fraud as the focal points for this study is driven by their substantial economic impact, pervasiveness, representativeness of broader fraud strategies, data availability, and alignment with industries actively seeking advanced detection solutions. By targeting these two critical domains, this research aims to make a significant contribution to the field of fraud detection with potential for wider applicability.

## State-of-the-art CNN model

CNNs mimic the human brain's cerebral cortex's intricate structure, exhibiting remarkable effectiveness. Their performance relies on a substantial training dataset, facilitating the development of a sophisticated model. Feature learning in this model is accomplished through the utilization of the backpropagation method and the gradient descent optimization algorithm. The system framework of the proposed method is visually presented in Fig. 1. The architectural design you have elucidated corresponds to a classification-oriented CNN. It encompasses diverse layers, including fully connected layers, batch normalization, rectified linear unit (ReLU) activation, dropout, softmax, and classification layers. In the subsequent discussion, we shall delve into the mathematical equations of each layer within this architecture. We first describe the structure of the art CNN model and then describe our proposed approach for credit card and auto insurance fraud detection.

### *Feature input layer*

This layer serves as the input to the neural network, representing the input features. Let's denote these input features as $x$, where $x \in R^{148}$. In this context, $R^{148}$ signifies the vector space of real numbers with a dimension of 148, indicating that there are 148 features included in the Vehicle Insurance Fraud Detection dataset.

### *Fully connected layer (500 neurons)*

The fully connected layer is essentially an artificial neural network that consists of multiple hidden layers connecting the input and output layers. These layers are composed of neurons, with each neuron assigned specific weights. The neurons in adjacent layers are interconnected through channels that have designated bias values. To achieve superior classification performance in convolutional neural networks, it is crucial to optimally adjust the bias and weight values of the fully connected layer during the training phase. The input to the fully connected layer is a vector, which represents the output from the previous layer.

$$k_{pj} = \sum_{i=1}^{m} x_i^{l-1} \mathbb{W}_{ji}^l + b_j^l \tag{1}$$

where

- $x_i$ is an input features, these are the outputs from the neurons of the previous layer (or the original input data for the first hidden layer in the network. In this context, each $x_i$ is a single feature from the set of features that are being input into the current neuron.
- $m$ is the total number of input features to the neuron $i$ in the current layer. If the current layer is the first hidden layer in the neural network, $m$ would be the number of features in the input dataset. If it is a deeper layer, $m$ is the number of neurons in the previous layer.
- $W_{ji}^l$ represents the weight for the connection from neuron $i$ in layer $l-1$ to neuro $j$ in layer $l$, and $b_j^l$ is the bias of neuron $j$ in layer $l$.
- Weighted Sum $k_{pj}$ denotes the sum of each input feature $x_i$ multiplied by its corresponding weight $W_{ji}$, plus the bias $b_j$. This sum is then usually passed through an activation function to produce the final output of the neuron (*Khedgaonkar, Singh & Raghuwanshi, 2021*; *Sindi et al., 2021*).

### *Batch normalization*

If the data is not normalized prior to training, our network might encounter obstacles, leading to a more arduous training process and a slower learning rate for the network. To address this problem, we employ the batch normalization technique, wherein normalization is applied to smaller data batches instead of the entire dataset at once. This strategy accelerates the preprocessing phase and allows for the utilization of higher learning rates, thereby facilitating smoother learning. During batch normalization, the means ($\mu_B$) and variances ($\sigma_B^2$) of each input layer are stabilized through the normalization step.

$$\mu_B = \frac{1}{m} \sum_{i=1}^{m} x_i \tag{2}$$

$$\sigma_B^2 = \frac{1}{m} \sum_{i=1}^{m} (x_i - \mu_B)^2 \tag{3}$$

where

- $B$ denotes the mini-batch of the size m of the whole training set;

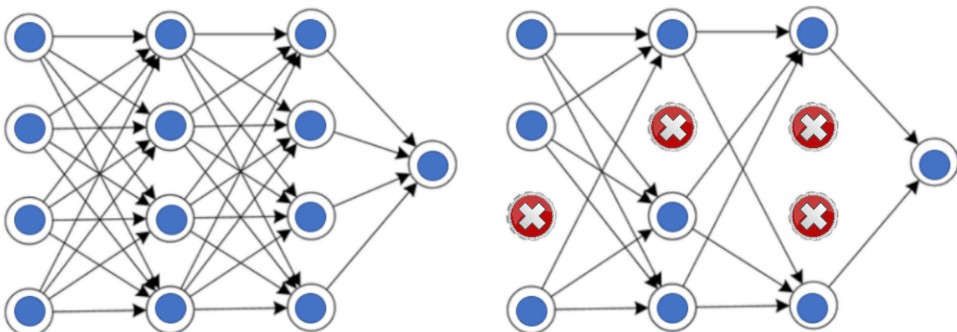

**Figure 2  The dropout layer.**

- $x_i$ is obtained by subtracting average value from $x$ and dividing by standard deviation. In order to prevent divide by zero, we add a small number $\varepsilon$ to the denominator.

$$x_i norm = \frac{x_i - \mu_B}{\sqrt{\sigma_B^2 - \varepsilon}} \tag{4}$$

- $\overline{x_i norm}$ is obtained *via* multiplying $x_i norm$ with a scale $\gamma$ and adding a shift $\beta$. Instead of using $x_i norm$ directly, $\overline{x_i norm}$ is utilized as the input for the non-linearity. $\gamma$ and $\beta$ are learned during the training process. $\overline{x_i norm}$ is defined as

$$\overline{x_i norm} = \gamma x_i norm + \beta. \tag{5}$$

### ReLU

ReLU and softmax functions are commonly utilized as activation functions in traditional CNN architectures. In the standard CNN framework, ReLU is applied after each convolution layer. Its primary objective is to introduce non-linearity by replacing negative values within the network with zeros, thus promoting sparsity. This characteristic enhances the network's sensitivity to subtle data variations. Mathematically, the ReLU function is defined as follows.

$$(x) = \begin{cases} 0 & for \ x < 0 \\ x & for \ x \geq 0 \end{cases}. \tag{6}$$

### Dropout layer (0.5)

Another crucial component of CNN is the Dropout layer. This layer acts as a mask, reducing the contributions of certain neurons to the subsequent layer while allowing other neurons to function, as depicted in Fig. 2. When a Dropout layer is applied to an input vector, some of its features are diminished, while certain hidden neurons are excluded in the case of a hidden layer. Dropout layers play a vital role in CNN training as they effectively mitigate the risk of overfitting the training data. In our research, Dropout layers with a threshold of 0.25 have been employed.

### SoftMax layer

The softmax activation function sends out the normalized form of the inputs. It accomplishes this by ensuring that all of its outputs are exactly equal to 1. The SoftMax Activation function provides the probability of a data point belonging to each individual class.

$$\sigma(\vec{z})_i = \frac{e^{z_i}}{\sum_{j=1}^{k} e^{z_j}} \tag{7}$$

where $\vec{z}$ isthe input vector, $e^{z_i}$ is the exponential function for input vector, $K$ is the number of classes, $e^{z_j}$ is the exponential function for output vector (*Sindi et al., 2021*).

### Classification layer

In a typical classification network, the final layer, known as the classification layer, is responsible for calculating the cross-entropy loss, which is crucial for both standard and weighted classification tasks involving distinct, non-overlapping classes. This layer generally comes after a softmax layer. During training, the classification layer utilizes the output from the softmax function to allocate each input to one of the K distinct classes. This allocation is based on the cross-entropy function, employing a 1-of-M coding scheme. The binary cross entropy loss function is generally used. It is defined as

$$Z_{binary} = -\frac{1}{m} \sum_{j=1}^{m} \left( y_j \log \hat{y}_j + (1 - y_j) \log(1 - \hat{y}_j) \right) \tag{8}$$

where $m$, $y_j \in [0,1]$, and $\hat{y}_j \in [0,1]$ denote the class number, the target value, and the predicted score, respectively (*Guo et al., 2023*).

## Feature extraction

Feature extraction initiates with a dataset comprising the primary features and employs them to produce supplementary features that are intended to be informative and devoid of redundancy. This procedure aids in the ability to generalize and potentially enhances the interpretation and performance scores for classification tasks.

Feature extraction, a fundamental preprocessing step in machine learning, involves converting dataset information into a more analytically comprehensible format. Traditionally, this task has relied on human expertise. However, deep learning models have revolutionized this field by automatically extracting features from raw data. Deep learning, especially through CNNs, autonomously processes and learns features from signals or images without human intervention.

In our study, we develop a CNN with a fully connected layer to distinguish between fraudulent and non-fraudulent cases. CNNs comprise multiple layers, including fully connected layers, pooling, and Batch Normalization, which play crucial roles. The fully connected and pooling layers primarily handle feature extraction from the input data. Nevertheless, increasing the number of layers in a network escalates its complexity and the risk of overfitting. To address this concern, our research focuses on training a model that effectively extracts features and classifies samples while maintaining a minimal layer structure to reduce complexity.

## High-performance filtering

High-performance filtering (HPF) is a methodology where the two most effective machine learning algorithms are chosen for integration into our ensemble learning framework. The primary goal of this HPF strategy is to identify the most suitable estimators that can be incorporated into the ensemble model to yield the final results. This approach focuses on optimizing the accuracy and consistency of the model's predictions by utilizing the best-performing algorithms in the ensemble configuration. However, it's important to note that if suboptimal accuracy is observed in half of the seven machine learning algorithms evaluated, the ensemble models may not surpass the performance of the base learners. To mitigate this risk and enhance robustness, a strategic selection of the top two algorithms. This selective approach not only addresses potential performance issues but also reduces the overall training time. The subsequent section of our study presents a detailed analysis of the feature extraction performance observed during our experiments. Having identified the top-performing machine learning algorithms through High-Performance Filtering, we then integrate these into our ensemble model to enhance overall performance. Specifically, the two most effective algorithms are selected for each dataset and incorporated into an ensemble voting classifier, employing a weighted average approach (soft voting).

Soft voting ensemble learning is an ensemble method that combines the predictions from multiple machine learning models to make a more robust final prediction. It considers the predicted probabilities for each class that each constituent model outputs and computes a weighted average of these probabilities. Specifically, a weighted average of the class probabilities is calculated using preset model weights to account for differences in each model's expected competence on the problem at hand.

In soft voting, multiple base models such as Support Vector Machines, K-Nearest Neighbors, Decision Trees *etc.* are trained independently on the same dataset. During prediction, each model generates probability estimates for class membership given a new data instance. For example, in a binary fraud classification application, each model may output the probability that the instance is fraudulent or legitimate. The individual probability outputs from all models are then averaged and weighted to determine the ensemble model's final probability estimates per class. The ensemble model ultimately classifies the instance into whichever class achieves the highest weighted probability.

Compared to relying on any single model, this soft voting approach allows multiple specialized models to contribute their complementary strengths in making more robust predictions. By aggregating models that are likely to make diverse errors, the ensemble can help cancel out biases, reduce variance, and enhance predictive performance beyond what any individual constituent model can achieve alone. The class with the highest weighted probability average is selected as the final prediction. The ability of ensembles formed by soft voting to make collective predictions leverages the differentiated skills of multiple models, thus improving generalization capacity and reducing sensitivity to noise *versus* individual models. By combining calibrated probability outputs across models, the soft voting ensemble can produce accurate and stable classifications. This is particularly valuable in applications like fraud detection systems, where the cost of diagnostic errors can be quite high (*Akyol et al., 2024*)

## Ensemble classifier

To construct an efficient model, ensemble models are formed by amalgamating base models. The ensemble model utilizes a combination of diverse learning algorithms to effectively address a classification or regression problem that cannot be adequately resolved by any single model in isolation. In this research, we used soft voting ensemble learning. Initially, we used training data to train basic models such as NB, SVM, DT, and K-NN as shown in Fig. 1. Following training, we used test data to evaluate our models' performance, with each model making an individual prediction. The ensemble learning approach incorporates the predictions of these models as supplementary inputs, enabling the ensemble model to learn and generate the final prediction collectively.

### Support vector machine

SVM, a supervised machine learning algorithm utilized for classification and regression (*Björklund, 2018*), operates by identifying the hyperplane that effectively distinguishes between classes for classification purposes. The algorithm strives to maximize the margin in order to position the hyperplane optimally. In the kernel trick approach, a kernel function is employed to transform the input space from a lower-dimensional to a higher-dimensional representation, proving particularly valuable in addressing non-linear separable problems.

### Decision tree

DT is a tree structure that resembles a flowchart, with each leaf node serving as a representation of the result and each interior node representing attribute (*Islam & Nahiduzzaman, 2022*). A root node is located at the very top of a decision tree. It becomes able to partition data based on a value of the attribute. A method of partitioning the tree recursively is with recursive partitioning. This form, which resembles a flowchart, aids in decision-making. It is a visualization in the form of a flowchart diagram that closely mimics human thought. So that, Decision trees are therefore easy to understand and interpret.

### Naive Bayes

The NB classifier is a probabilistic model for machine learning that is used for classification problems of two-class (binary) and multi-class (*Majeed, Abdullah & Mushtaq, 2021*). The NB classifier relies on the Bayes theorem:

$$P(C|Z_1,\ldots,C|Z_n) = \frac{P(C)\pi_{h=1}^n P(Z_h|C)}{\pi_{h=1}^n P(Z_h)} \tag{9}$$

where, $P(C|Z_1,\ldots,C|Z_n)$ is the conditional probability (CP) of class (C) given features ($Z_1$, $Z_2, \ldots, Z_n$); $P(C)$ is the prior probability of C; $P(Z_h|C)$ is the CP of $Z_h$ ($h=1,2,3,\ldots,n$) given C ; and $P(Z_h)$ is the prior probability of $Z_h$.

### K-nearest neighbors classifier

In several fields, KNN is applied as a classification or regression algorithm. In either case, the training set is expanded using the additional input data from the nearby samples. On the other hand, the results vary depending on whether the KNN algorithm is applied to regression or classification. If the KNN algorithm is employed to classify, the object is classified by a common vote of its neighbors. Otherwise, if the KNN algorithm is employed

for regression, the output accuracy depends on determining the average value of the nearest neighbors for the same object (*Abdel-Kader, El-Sayad & Rizk, 2021*). The KNN algorithm stands out as the optimal choice for developing a classification system without making any assumptions about the function's shape. KNN endeavors to classify new objects by considering their attributes in relation to the training samples. The nearest neighbors are identified by evaluating variables through a standardized Euclidean distance calculation. In Euclidean geometry, the distance between two points is determined by the following equation.

$$D = \sqrt{(p_1 - n_1)^2 + (p_2 - n_2)^2 + \ldots + (p_i - n_i)^2} \tag{10}$$

where $p$ is a predictor variable and $n$ is a new point.

# PROPOSED METHOD

This section presents a comprehensive five-stage process for the proposed Convolutional Neural Network-Machine Learning (CNN-ML) methodology, designed specifically for binary classification of fraud, as depicted in Fig. 1. The initial stage involves the introduction of a dataset, consisting of two distinct public datasets, into the system. Subsequently, the dataset undergoes preprocessing to ensure its compatibility with the input requirements of the CNN model, representing the second stage. The third stage focuses on the training, validation, and testing of a state-of-the-art CNN model to evaluate its effectiveness in classifying fraud instances. In the fourth stage, the features extracted by the CNN models undergo further processing using machine learning algorithms. The final stage entails testing the trained models and assessing their performance using established classification metrics such as accuracy, precision, recall, F1-score, and AUC. Each of these stages in the proposed approach is elaborated upon in subsequent sections.

## Data preprocessing

Initially, it was observed that the original datasets for Vehicle Insurance and European credit card transactions exhibited a discrepancy in the number of instances between the fraud and non-fraud categories, as illustrated in Table 1. This imbalance within the database posed challenges for developing an effective predictive model, particularly for the minority category representing fraud in the Vehicle Insurance and European credit card transactions datasets. To address this issue, we employed oversampling technique (*Buda, Maki & Mazurowski, 2018*), which involved randomly replicating instances from the minority category to match the number of instances in the majority category, as demonstrated in Table 1.

Secondly, the data is categorized into two types: fraud and non-fraud, where the majority represents normal cases and the minority, abnormal. The dataset features are of two kinds: categorical and numerical. Categorical features further include both ordered and unordered types. These datasets have categorical features, some of which are not sequentially ordered. For example, the Vehicle Insurance dataset includes the marital status of users. Since computers only process numerical data, these categorical features must be

digitally converted. For non-sequential categorical features, we use One-Hot Encoding. For instance, the gender feature, with two categories, is represented as one-hot vectors: male = (1) and female = (0). However, some features exhibit a sequential order, like day of week, which includes seven categories: Saturday to Friday. For these sequential features, One-Hot Encoding is unsuitable due to the inherent order among categories. Instead, we apply sequential encoding, assigning distinct natural numbers to each category. For instance, the Vehicle Insurance dataset includes the marital status of users. Since computers only process numerical data, these categorical features must be digitally converted. For non-sequential categorical features, we use One-Hot Encoding. For instance, the gender feature, with two categories, is represented as one-hot vectors: male = (1) and female = (0). However, some features exhibit a sequential order, like day of week, which includes seven categories: Saturday to Friday. For these sequential features, One-Hot Encoding is unsuitable due to the inherent order among categories. Instead, we apply sequential encoding, assigning distinct natural numbers to each category. For example, days of the week are mapped as Saturday = (1), Sunday = (2), Monday = (3), and so on through Friday = (7).

## Proposed CNN model

The objective of this study is to employ a newly developed CNN model to classify the datasets into two distinct categories: fraud and non-fraud. The architecture of this model comprises a total of 12 layers, encompassing a feature input layer, convolution layers with Rectified Linear Unit (ReLU) activation functions, batch normalization, maximum pooling, two fully connected layers, and a softmax activation layer for the final output. The detailed structure of the model can be illustrated in Fig. 1, while specific parameter values for each layer are provided in Table 2.

## Machine learning algorithms

The machine learning algorithms leverage the extracted features from the feature layers of the CNN model to perform classifications. The specific layer of the trained CNN model is selected and employed for feature extraction. our approach utilizes the fc_3 layer from the CNN model, as outlined in Table 1. Subsequently, the classification phase involves the processing of the extracted features from this single layer. For classification purposes, we employ the NB, SVM, DT, and KNN algorithms as classifiers, as depicted in Fig. 1.

## Performance metrics

When evaluating the effectiveness of machine learning and deep learning models, various metrics are utilized, with specific learning tasks often requiring particular metrics for emphasis (*Shaukat et al., 2020*). For this study, we consider the following metrics as essential and provide detailed explanations below.

Accuracy, which is defined as the ratio of correctly predicted instances to the total number of predictions, serves as a fundamental metric for evaluating classification model precision. A higher accuracy value signifies a more precise classification model, making it a desirable outcome (*Deng et al., 2016*).

Another crucial metric is the F1-score, which harmonizes the precision and recall of a model. This metric proves particularly valuable in scenarios where there is a skewed class

**Table 2 Classification results of testing dataset for CNN model of Vehicle Insurance fraud dataset.**

| Deep CNN | Accuracy% | Sensitivity% | Specificity% | F1-Score% | Precision% |
|---|---|---|---|---|---|
| Proposed CNN | 64.18 | 33.85 | 94.65 | 48.65 | 86.42 |

distribution, such as a significant discrepancy in sample sizes across classes. F1 score enables a comprehensive assessment of model performance by providing a balanced measure between precision and recall. A higher F1-score indicates superior model performance compared to other models (*Shaukat et al., 2020*). These metrics are computed as

$$Accuracy = \frac{TP + TN}{TP + TN + FN + FP}, \tag{11}$$

$$precision = \frac{TP}{TP + FP}, \tag{12}$$

$$Recall = TPR = \frac{TP}{TP + FN} \tag{13}$$

$$F1 - score = TPR = \frac{2TP}{2TP + FP + FN} \tag{14}$$

$$Specificity = TNR = \frac{TN}{TN + FP}. \tag{15}$$

True positive (TP) denotes the count of positively labeled samples that the model has accurately identified as positive. Conversely, true negative (TN) represents the quantity of samples correctly classified as negative by the model, referring to instances genuinely belonging to the negative class. False positive (FP) indicates the instances that are incorrectly labeled as positive by the model, despite being negative. on the other hand, false negative (FN) is the count of positive samples erroneously classified as negative (*Shaukat et al., 2020*).

## RESULTS AND DISCUSSION OF THE PROPOSED CNN MODEL AND ENSEMBLE ML CLASSIFIER

The main aim of this study is to utilize a recently developed CNN model for the classification of two imbalanced datasets, namely Vehicle Insurance Fraud and European Credit Card Transactions Fraud, into binary categories: fraud and non-fraud. The findings of this study highlight the CNN model's strong ability to handle large, complex and imbalance datasets for fraud detection, as demonstrated by its performance on the European Credit Card Transactions dataset. However, it should be noted that the effectiveness of the CNN model is somewhat constrained when applied to smaller datasets like the Vehicle Insurance Fraud dataset.

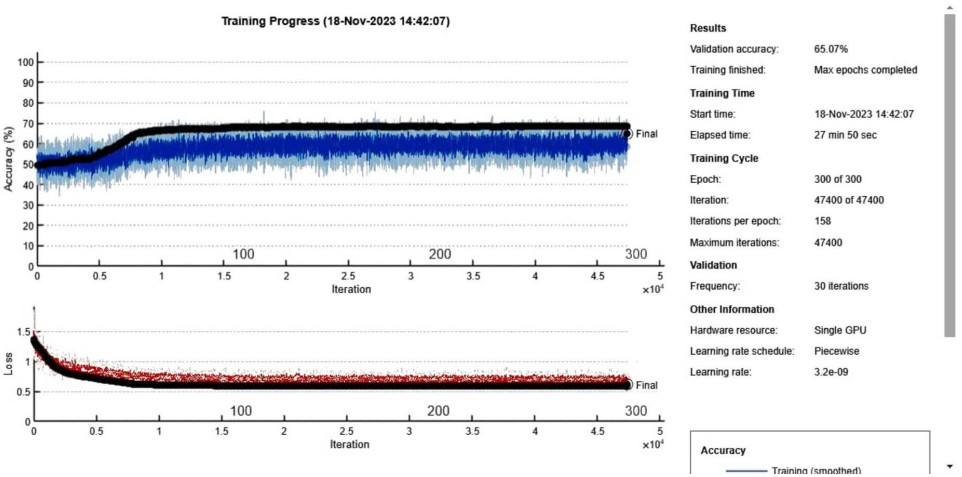

**Figure 3** Accuracy and loss curves for proposed CNN model for Vehicle Insurance fraud dataset.

## Performance evaluation for auto insurance fraud dataset

Initially, the auto insurance fraud oversampled dataset was divided into three subsets: 70% for training, 15% for validation, and 15% for testing purposes. The proposed CNN model for the Vehicle Insurance Fraud dataset was trained over 300 epochs, using a mini-batch size of 15 and a total of 47,400 iterations, which took approximately 27 min and 50 s. The Adam solver was employed with an initial learning rate of 0.00001 for optimization. Figure 3 presents the training and validation graphs, showing the progression of loss values and accuracy for the proposed CNN model. During training, the model achieved an accuracy of 60% with a corresponding loss of 0.7, while the validation accuracy was also 60% with a loss of 0.7%. Specific hyperparameters used for training the CNN model, such as learning rate, batch size, number of epochs, *etc.* This will allow reproducibility of our results.

The performance metrics obtained from the CNN model are presented in Table 2 and Fig. 4, both in tabular format and as a visual graphical bar chart. These results indicate that the CNN model did not accurately classify the Vehicle Insurance Fraud dataset, likely due to the limited amount of data available. Deep learning models generally require a larger dataset to achieve better performance.

Table 3 and Fig. 5 present the performance evaluation of seven machine learning (ML) techniques used to classify features extracted from the CNN model. The results demonstrate the accurate classification of features achieved by the ML techniques. Among the methods examined, Knn1 emerged as the most effective ML technique, achieving a high accuracy of 96.62%. This was accomplished by extracting features from the performance metrics of the Vehicle Insurance Fraud dataset using the CNN model and subsequently classifying them with Knn1. To further enhance the classification accuracy, a soft voting ensemble-based approach was implemented for the final detection. The CNN-ensemble ML model's robustness was evaluated using confusion matrices for each dataset, considering metrics such as accuracy, precision, sensitivity, F1-score, and AUC. In the context of the

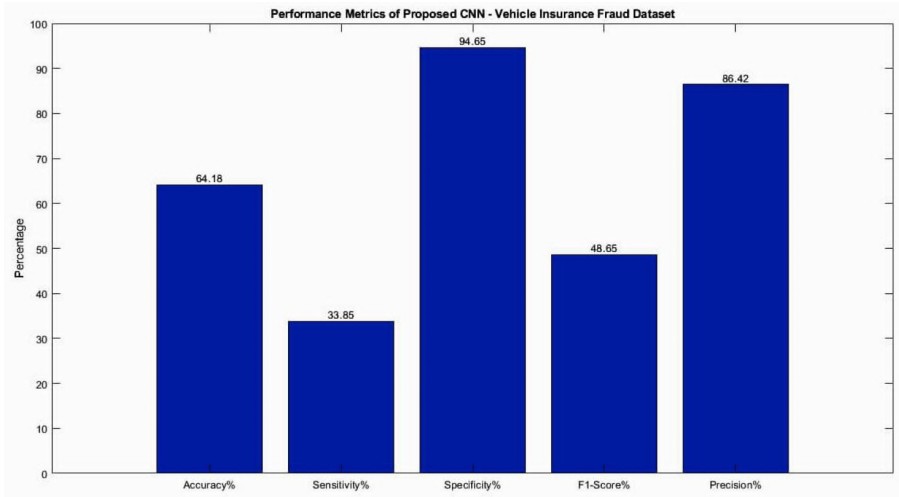

**Figure 4** Accuracy performance analysis for testing dataset for CNN model of Vehicle Insurance Fraud dataset.

**Table 3** Performance metrics obtained from ensemble model of Testing dataset of Vehicle Insurance Fraud dataset.

| Classifiers | Accuracy% | Sensitivity% | Specificity% | Precision% | F1_score |
|---|---|---|---|---|---|
| DT | 96.53 | 93.05 | 100 | 100 | 96.40 |
| Knn5 | 90.91 | 81.77 | 100 | 100 | 89.97 |
| Knn3 | 93.38 | 86.72 | 100 | 100 | 92.89 |
| Knn1 | 96.62 | 93.22 | 100 | 100 | 96.49 |
| GSVM | 69.98 | 54.20 | 85.66 | 78.98 | 6,429 |
| LSVM | 68.75 | 50.29 | 87.10 | 79.50 | 61.61 |
| NB | 68.29 | 45.00 | 91.44 | 83.94 | 58.59 |
| Ensembles | 98.86 | 97.72 | 100 | 100 | 98.85 |

finance sector, maximizing sensitivity is crucial as it ensures timely and accurate detection of fraudulent digital payments.

Figure 6 displays the confusion matrix for the CNN-ensemble ML architecture applied to the Vehicle Insurance Fraud dataset, which demonstrated a lower misclassification The misclassification errors for Non-Fraud and Fraud were found to be 0 and 66, respectively. The accuracy values of the DT, Knn5, Knn3, Knn1, GSVM, LSVM, NB, and ensemble models were determined to be 96.53%, 90.91%, 93.38%, 96.62%, 69.98%, 68.75%, 68.29%, and 98.86%, respectively, as presented in Table 3, showcasing the classification performance metrics of all the ML models.

In ML and DL, the AUC-ROC curve holds significant importance in evaluating the performance of classification tasks, particularly in the context of imbalanced datasets. This curve serves as a valuable metric for assessing the predictive accuracy of a model. The ROC curve represents the probability of correct classification for different classes, with the

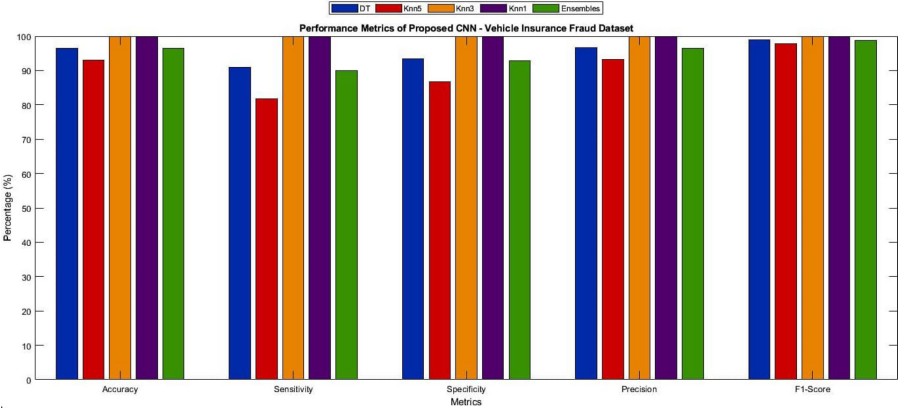

**Figure 5** **Performance metrics obtained from ensemble model for different ML algorithms of testing dataset of Vehicle Insurance Fraud dataset.**

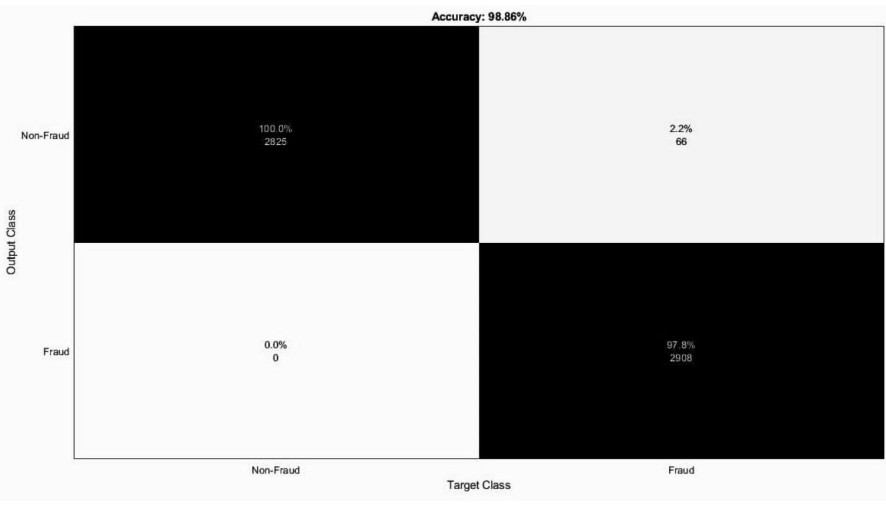

**Figure 6** **Confusion matrix for ensemble soft voting classifier for Vehicle Insurance Fraud dataset.**

$X$-axis representing the False Positive Rate (FPR) and the $Y$-axis representing the True Positive Rate (TPR).

Regarding the CNN-ensemble ML architectures, Fig. 7 illustrates the ROC curves and corresponding AUC values for different CNN-ensemble ML configurations. The CNN-ensemble ML model proposed in this study exhibits an impressive AUC value of 96.5%, indicating its exceptional predictive precision for the Vehicle Insurance Fraud dataset.

## Comparison with previous studies for auto insurance fraud dataset

This section demonstrates the superior performance of our ensemble model in comparison to previous endeavors within the same domain. Table 4 provides a comparative analysis of

Ming et al. (2024), *PeerJ Comput. Sci.*, DOI 10.7717/peerj-cs.2088

our proposed approach against existing methodologies in classifying the Vehicle Insurance Fraud dataset.

Several studies have investigated deep learning approaches for auto insurance fraud detection, focusing on imbalanced datasets. *Xu et al. (2023)* developed Deep Boosted Decision Trees utilizing gradient boosting and neural networks, with a specific emphasis on maximizing AUC performance. They evaluated an auto insurance fraud dataset but did not provide comprehensive accuracy or other statistical measures. More relevantly, *Xia, Zhou & Zhang (2022)* proposed a CNN-LSTM model for auto insurance fraud detection, aiming to reduce the complexity of expert-driven feature engineering. By automatically learning feature representations, their approach achieved an accuracy of 89.6% and precision of 90.7% on a standard insurance dataset.

The first column of the table lists prior studies, while the second column outlines the methodologies employed in each study, which could entail either the introduction of a novel model by the authors or the application of cutting-edge models. The third column provides details about the datasets utilized in each study. To facilitate a comprehensive assessment and identification of the most effective strategies, the table highlights the best method and the highest achievement for each study.

According to *Table 4*, our approach combining CNN with ensemble machine learning techniques (KNN1 and DT) outperforms some of the state-of-the-art models. These methods excel in extracting more robust and distinct deep features, leading to optimal classification outcomes. Additionally, to address the dataset's imbalance, an oversampling technique is employed during network training. Despite the utilization of different datasets, our proposed CNN consistently delivers superior performance compared to other studies that incorporate dual classification to tackle the similar issue.

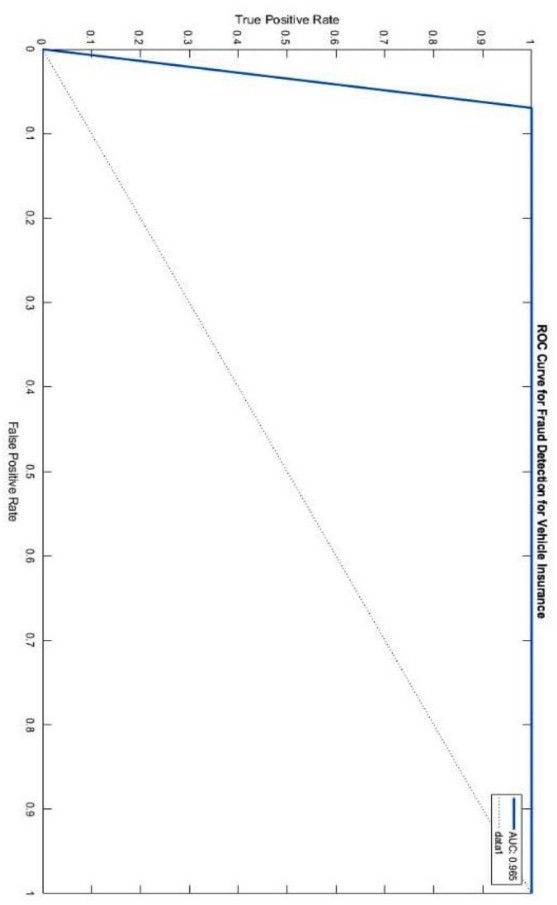

**Figure 7** ROC for ensemble soft voting classifier for Vehicle Insurance Fraud dataset.

**Table 4  Comparison of the proposed method with previous studies.**

| Previous studies | Model or Algorithm | Dataset | Number of Classes | The class imbalance problem and how it has been resolved in the literature | Accuracy% | Precision% | Recall% | F1-score% | AUC% |
|---|---|---|---|---|---|---|---|---|---|
| *Xu et al. (2023)* | Deep Boosting Decision Trees (DBDT), which combines gradient boosting and neural networks | Vehicle Insurance Fraud dataset | 2 | The research emphasizes the importance of AUC maximization for balanced data performance | | | | | 81.236% |
| *Xia, Zhou & Zhang (2022)* | Proposed combination of CNN and LSTM. The model was able to reduce the need for complex feature extraction processes that often rely on domain experts in traditional machine learning algorithms. | Vehicle Insurance Fraud dataset | 2 | Used SMOTE | 89.60% | 90.70% | 89.60% | | |
| Proposed method | CNN with ensemble Machine learning models | European credit card transactions Fraud | 2 | Oversampling method | 98.86 | 100% | 97.72% | 98.85% | 100% |

## Performance evaluation for European credit card transactions fraud dataset

The proposed CNN model for the European credit card transactions Fraud dataset utilized an oversampled dataset. The dataset was divided into 70% for training, 15% for validation, and 15% for testing. The model was trained for 300 epochs using a mini-batch size of 15, resulting in 129,900 iterations. The training process took approximately 312 min and 29 s. The Adam solver was employed using an initial learning rate of 0.00001. Figure 8 presents the training and validation graphs, showcasing the loss values and accuracy of the proposed CNN model. The training accuracy reached 94.54% with a loss of 0.2, while the validation accuracy also achieved 94.54% with a loss of 0.2%. The performance metrics obtained from the CNN model are presented in Table 5 and visualizes in Fig. 9, providing a comprehensive overview of the model's effectiveness in accurately classifying European credit card transactions Fraud. Deep learning algorithms, such as CNN, benefit from large amounts of training data, and the European credit card transactions Fraud dataset contains substantial data suitable for such purposes. Furthermore, Table 6 and Fig. 10 present the performance of seven machine learning techniques employed to classify the features of the CNN model, offering a comparative analysis.

The results demonstrate that the ML techniques accurately classified the features extracted from each CNN model. Among the methods, Knn1 emerged as the strongest ML technique for classification, achieving the highest accuracy of 99.97%. This was accomplished by extracting features from the performance metrics of the European credit card transactions Fraud dataset using the CNN model and classifying them with Knn1. Furthermore, Fig. 11 presents the confusion matrix for the CNN-ensemble ML architecture applied to the European credit card transactions Fraud dataset, which exhibited a lower misclassification error rate. The misclassification errors for Non-Fraud and Fraud were 0 and 15, respectively. The accuracy values for the DT, Knn5, Knn3, Knn1, GSVM, LSVM, NB, and ensemble models were 99.97, 99.93, 99.95, 99.97, 94.64, 94.92, 94.50, and 99.99, respectively, as shown in Table 5, which presents the classification performance measures for all ML models.

Regarding the CNN-ensemble ML architectures, Fig. 12 display the ROC curves and corresponding AUC values for the CNN-ensemble ML configurations. The CNN-ensemble ML model, as the proposed model, demonstrates an AUC value of 100%, indicating its exceptional predictive precision for European credit card transactions.

## Comparison with previous studies for European credit card transactions fraud dataset

This section showcases the superior performance of our ensemble model compared to previous endeavors in the field. Table 7 presents a comparative analysis of our proposed approach in classifying the European credit card transactions Fraud dataset against existing methodologies.

Several studies have explored machine learning techniques for credit card fraud detection. For instance, *Lei et al. (2023)* utilized distributed deep neural networks and achieve a remarkable accuracy of 99.94%. *Xu et al. (2023)* proposed Deep Boosted Decision Trees

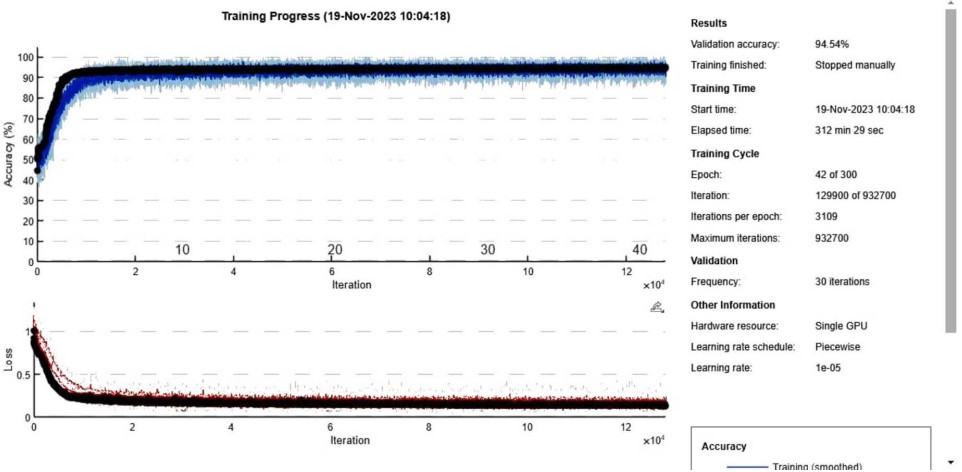

**Figure 8 Accuracy and loss curves for proposed CNN model for European credit card transactions Fraud dataset.**

**Table 5 Classification results of Testing dataset of European credit card transactions Fraud dataset.**

| Deep CNN | Accuracy% | Sensitivity% | Specificity% | F1-Score% | Precision% |
|---|---|---|---|---|---|
| Proposed CNN | 94.50 | 99.06 | 89.98 | 94.72 | 90.74 |

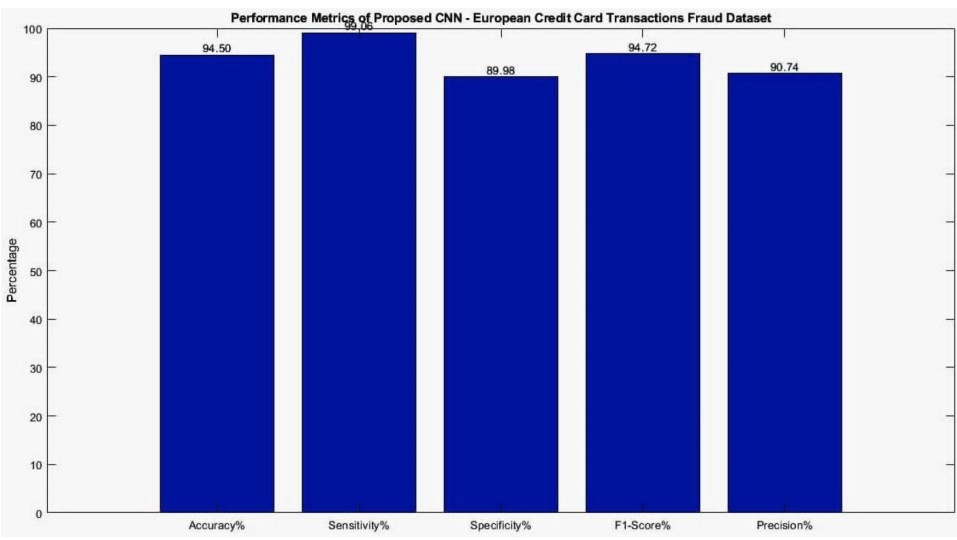

**Figure 9 Performance metrics analysis for Testing dataset for CNN model of European credit card transactions Fraud dataset.**

that combine neural networks and gradient boosting, with a specific focus on optimizing the AUC metric and attaining an AUC of 81.24%. *Sharma, Singh & Chandra (2022)* design a CNN-based architecture incorporating GANs and SMOTE for oversampling, achieving an

**Table 6** Performance metrics obtained from ensemble model of Testing Dataset of European credit card transactions Fraud dataset.

| Classifiers | Accuracy% | Sensitivity% | Specificity% | Precision% | F1_score |
| --- | --- | --- | --- | --- | --- |
| DT | 99.97 | 99.93 | 100 | 100 | 99.97 |
| Knn5 | 99.93 | 99.86 | 100 | 100 | 99.93 |
| Knn3 | 99.95 | 99.89 | 100 | 100 | 99.95 |
| Knn1 | 99.97 | 99.94 | 100 | 100 | 99.97 |
| GSVM | 94.64 | 98.85 | 90.46 | 91.13 | 94.83 |
| LSVM | 94.92 | 98.79 | 91.09 | 91.66 | 95.09 |
| NB | 94.50 | 98.39 | 90.64 | 91.25 | 94.69 |
| Ensembles | 99.99 | 99.97 | 100 | 100 | 99.99 |

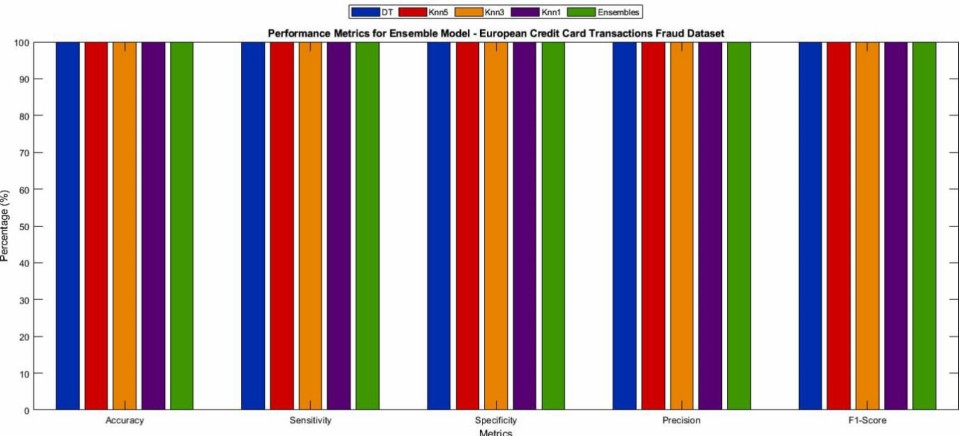

**Figure 10** Performance metrics obtained from ensemble model for different ML algorithms of testing dataset of European credit card transactions Fraud dataset.

85% precision; however, they do not evaluate AUC. More closely aligned with our proposed approach, *Fanai & Abbasimehr (2023)* develop a two-stage model involving an autoencoder for dimensionality reduction followed by deep learning classifiers. Their best-performing configuration, employing CNN and RNN, achieves an F1- score of 83.72%.

To further enhance the identification of credit card fraud in imbalanced datasets, we propose an integrated framework that combines a CNN-based feature extractor with ensemble machine learning classifiers. Our model incorporates targeted oversampling to balance the class distributions, enabling the classifiers to better discern the characteristics of the minority positive class. We conduct a comprehensive empirical evaluation using stratified sampling and metrics such as precision, recall, and AUC, specifically designed for handling data skewness. The objective of our proposed system is to achieve nearly 100% precision, ensuring reliable fraud alerts, while simultaneously maximizing the detection rate. Through the utilization of deep feature representations and intelligent fusion of multiple classifiers, our technique outperforms previous benchmarks in terms of accuracy, F1-score, and AUC.

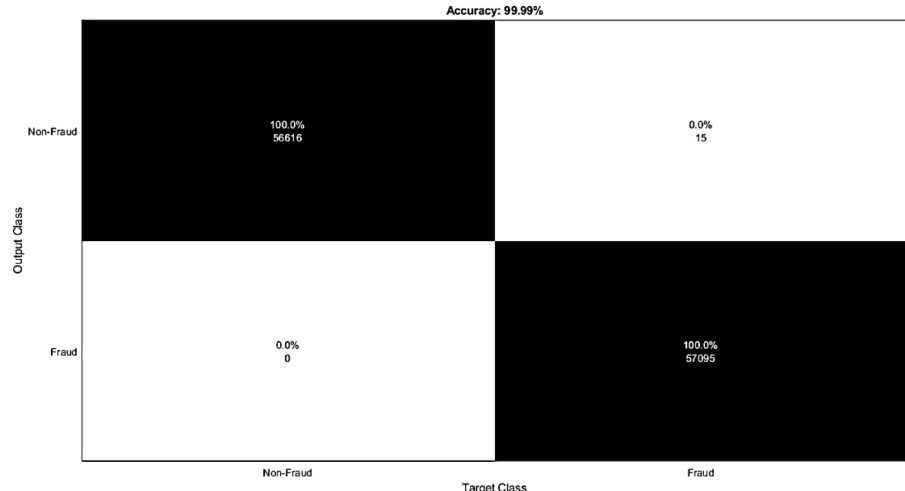

**Figure 11 Confusion matrix for ensemble soft voting classifier for European credit card transactions Fraud dataset.**

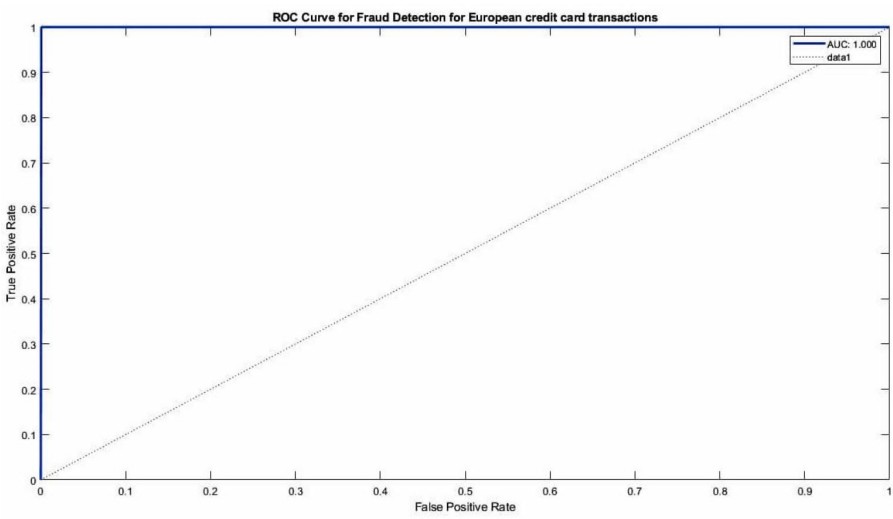

**Figure 12 ROC for ensemble soft voting classifier for European credit card transactions Fraud dataset.**

## CONCLUSION

This study introduces a novel and powerful approach to enhance the accuracy of fraud detection for auto insurance fraud and credit cards by harnessing the synergy between deep learning and four machine learning algorithms. The method of using deep features from CNNs as inputs to machine learning models represents a significant advancement in the field, offering enhanced detection capabilities while maintaining the necessary transparency for practical application in various sectors, including finance and auto insurance.

**Table 7  Comparison between the proposed model and previous studies.**

| Previous studies | Model or Algorithm | Dataset | Number of Classes | The class imbalance problem and how it has been resolved in the literature | Accuracy% | Precision% | Recall% | F1-score% | AUC% |
|---|---|---|---|---|---|---|---|---|---|
| *Lei et al. (2023)* | Distributed deep neural network. | European credit card transactions Fraud | 2 | Used the FedAvg aggregation algorithm | 99.94% | 99.965% | 99.98% | 99.971% | – |
| *Xu et al. (2023)* | Deep Boosting Decision Trees (DBDT), which combines gradient boosting and neural networks. | European credit card transactions Fraud | 2 | The research emphasizes the importance of AUC maximization for balanced data performance | | | | | 81.24% |
| *Sharma, Singh & Chandra (2022)* | Proposed a CNN architecture consists of fully connected neural network classifier with the following structure: Input layer, 2 Hidden layers with 128 neurons each, Output layer and Sigmoid activation function. | European credit card transactions Fraud | 2 | Hybrid generative model has been developed by combining the advantages of both GANs and SMOTE for oversampling | | 85% | 80% | 81.18% | |
| *Fanai & Abbasimehr (2023)* | Proposed a two-stage framework comprised of a deep auto encoder (AE) model as the dimensionality reduction technique, and three deep learning-based classifiers including DNN, RNN, and CNN-RNN to improve fraud detection accuracy. The best performing model (AE-CNN-RNN) achieved. | European credit card transactions Fraud | 2 | Proposed a framework comprised of A deep autoencoder neural network with multiple encoder and decoder layers is trained on the original imbalanced dataset. The trained autoencoder model is used to transform the original data into a lower dimensional representation by passing it through the encoder part and extracting the code layer features | | 96.77% | 73.77% | 83.72% | |
| *Alharbi et al. (2022)* | This paper uses a text2IMG conversion technique to transform transaction data into images, and then applies a 16-layer CNN architecture for fraud classification. The text2IMG conversion method proposed in the attached paper to represent transaction data as images is a novel aspect of their work | European credit card transactions Fraud | 2 | | 99.87% | | 51.22 | 57.8% | |
| *Mizher & Nassif (2023)* | | | | | 99.7% | 99% | 99% | 99% | 95% |
| Proposed method | CNN with ensemble Machine learning models | European credit card transactions Fraud | 2 | Oversampling method | 99.99% | 100% | 99.97% | 99.99% | 100% |

Our findings indicate that the proposed model demonstrates superior performance compared to previous studies across various types of fraud datasets, including auto insurance and credit cards. Notably, as the volume of data increases, the model maintains its effectiveness, ensuring consistent and reliable performance in diverse scenarios. For auto insurance fraud detection, our proposed model achieved the highest results, with an accuracy of 98.86%, precision of 100%, recall of 97.72%, F1-score of 98.85, and an AUC of 100%. Similarly, in the case of credit card fraud detection, our proposed model achieved exceptional outcomes, with an accuracy of 99.99%, precision of 100%, recall of 99.97%, F1-score of 100%, and an AUC of 100.

### Funding

The authors received no funding for this work.

### Competing Interests

The authors declare there are no competing interests.

### Author Contributions

- Ruixing Ming conceived and designed the experiments, analyzed the data, prepared figures and/or tables, and approved the final draft.
- Osama Abdelrahman conceived and designed the experiments, analyzed the data, performed the computation work, prepared figures and/or tables, authored or reviewed drafts of the article, and approved the final draft.
- Nisreen Innab performed the experiments, authored or reviewed drafts of the article, and approved the final draft.
- Mohamed Hanafy Kotb Ibrahim performed the experiments, performed the computation work, authored or reviewed drafts of the article, and approved the final draft.

### Data Availability

  The vehicle insurance fraud data is available at Kaggle:
  https://www.kaggle.com/datasets/shivamb/vehicle-claim-fraud-detection.
  The credit card fraud data is available at Kaggle:
  https://www.kaggle.com/datasets/mlg-ulb/creditcardfraud.
  The code is available in the Supplemental Files.

### Supplemental Information

Supplemental information for this article can be found online at http://dx.doi.org/10.7717/peerj-cs.2088#supplemental-information.

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
