# Peer review of "Enhancing fraud detection in auto insurance and credit card transactions: a novel approach integrating CNNs and machine learning algorithms"

_PeerJ Computer Science, doi:10.7717/peerj-cs.2088_

## Round 0.1 · original submission · Major Revisions

I have received reviews of your manuscript from scholars who are experts on the cited topic. They find the topic interesting; however, several concerns regarding experimental results and comparisons with current approaches must be addressed. These issues require a major revision. Please refer to the reviewers’ comments at the end of this letter; you will see that they advise you to revise your manuscript. If you are prepared to undertake the work required, I would be pleased to reconsider my decision. Please submit a list of changes or a rebuttal against each point that is being raised when you submit your revised manuscript.

Thank you for considering PeerJ Computer Science for the publication of your research.

With kind regards,

Reviewer 1 ·

Basic reporting

The manuscript introduces an AI-based approach to combating fraud in auto insurance and credit card transactions by leveraging Convolutional Neural Networks (CNNs) combined with traditional machine learning algorithms such as SVM, KNN, NB, and DT. This approach aims to enhance the accuracy and efficiency of fraud detection systems through the utilization of deep features extracted by CNNs. I have some significant concerns about this paper. I will discuss them as follows.

Experimental design

Selection of Fraudulent Activities Domains:
The manuscript targets two specific areas of fraud detection: auto insurance and credit card transactions. However, the rationale behind selecting these two domains over others is not clearly articulated. People often only target one specific domain. It would be beneficial for the authors to provide specific reasoning or evidence that underscores the significance of auto insurance and credit card fraud over other forms of fraud, elucidating why these particular areas(instead of others) were chosen for this study's focus.

Innovation and Comparison with Previous Works:
The integration of CNNs with machine learning algorithms is an approach that has been explored in prior research. The manuscript needs to provide a more comprehensive discussion on how the proposed model distinguishes itself from existing methodologies. Specifically, a critical analysis of the model to the study referenced (https://www.mdpi.com/2079-9292/11/5/756) and other relevant literature is necessary. This comparison should highlight the novel aspects of the proposed approach, its advantages, and potential improvements in fraud detection performance.

Validity of the findings

Please add experiments to compare this work with some other prior works which also integrate CNNs with machine learning algorithms.

Additional comments

1. Introduction Writing: The introduction currently encompasses a broad overview of fraudulent activities without a concise focus. Reducing this section to one or two paragraphs that succinctly outline the background of fraud, specifically in the chosen domains, would streamline the presentation and help clarify the motivation behind the research. Furthermore, a more detailed delineation of how this work diverges from other studies combining CNNs with machine learning algorithms is required. Such differentiation is crucial to understanding the unique contribution of this research. Also, in lines 104-116, please highlight concrete numbers in your evaluation results to show the good performance of your tool.

2. Please add necessary references throughout this paper.

Reviewer 2 ·

Basic reporting

• The basic reporting of the manuscript is clear. The motivation could be enhanced. The English language used is clear and the manuscript is quite easy to follow. The structure of the paper conforms to PeerJ standards. Figures are labeled correctly and in high quality. There are still some more profound works that are not reviewed in this work. The paper should give depth analysis.

Experimental design

The methodology section presents only the theoretical explanation of CNN model and machine learning models. The experimental setup can be improved. The dataset used in this research has been taken from the Kaggle dataset repository. It is also suggested to work on the different categories of datasets that consist of different features.

Validity of the findings

It looks that some significant findings were observed using Convolutional Neural Networks (CNNs) with SVM, KNN, NB, and DT algorithms. However, the findings can be evaluated after clarifying the models.

Additional comments

More recent papers should be reviewed. There are still some more profound works that are not considered in this work. The writing of this manuscript needs further improvements. There are some grammatical mistakes.

---

## Round 0.2 · accepted · Accept

I am pleased to inform you that your work has now been accepted for publication in PeerJ Computer Science.

Please be advised that you are not permitted to add or remove authors or references post-acceptance, regardless of the reviewers' request(s).

Thank you for submitting your work to this journal. On behalf of the Editors of PeerJ Computer Science, we look forward to your continued contributions to the Journal.

With kind regards,

Reviewer 2 ·

Basic reporting

The manuscript has been revised as per the reviewer's suggestions and it can be considered for publication.

Experimental design

as above

Validity of the findings

as above

Additional comments

as above